# A Mathematical Model of Plane-Parallel Movement of the Tractor Aggregate Modular Type

**Volodymyr Bulgakov** [1], **Aivars Aboltins** [2] , **Semjons Ivanovs** [2,*], **Ivan Holovach** [1], **Volodymyr Nadykto** [3] **and Hristo Beloev** [4]

1   Department of Mechanics, Faculty of Construction and Design, National University of Life and Environmental Sciences of Ukraine, 03041 Kyiv, Ukraine; vbulgakov@meta.ua (V.B.); holovach.iv@gmail.com (I.H.)

2   Institute of Agricultural Machinery, Faculty of Engineering, Latvia University of Life Sciences and Technologies, Cakstes Blvd. 5, LV-3001 Jelgava, Latvia; aivars.aboltins@inbox.lv

3   Department of Machine-Using in Agriculture, Dmytro Motornyi Tavria State Agrotechnological University, 18, Khmelnytskiy Ave., 72312 Melitopol, Ukraine; volodymyrnvt@meta.ua

4   Department of Agricultural Machinery, University of Ruse "Angel Kanchev", Studentska 8, 7017 Ruse, Bulgaria; hbeloev@uni-ruse.bg

*   Correspondence: semjons@apollo.lv; Tel.: +371-29-403-708

**Abstract:** The new machine-and-tractor aggregate of a modular type, developed by us, consisting of the power and the technological modules, can significantly increase the efficiency of using the tractor when it is aggregated with a five-body plough. The new design solution for connecting these modules in a horizontal plane through a damper, consisting of a hydraulic cylinder and a throttle, allows significant increasing of the movement stability of this aggregate in a transverse-horizontal plane. To test the efficient operation of the proposed design, as well as to determine the kinematic and design parameters that provide the required level of stable movement of this modular machine-and-tractor aggregate, we carried out theoretical and experimental field investigations. For this, a new mathematical model of plane-parallel movement of the machine-and-tractor aggregate of this type was built. It was found that a change in the operating speed of this aggregate during ploughing from 1.0 to 3.0 m·s$^{-1}$ does not lead to a deterioration in the stability of the movement of either the technological or, especially, the power modules. The delay in the reaction of the power module of the machine-and-tractor aggregate of the modular type is practically invariant with respect to the change in the mode of movement of this aggregate within the range 1–3 m·s$^{-1}$. It was also found that the values of the tire slip resistance coefficients of the wheels of the power module do not have a noticeable impact upon the development of fluctuations of the disturbing moment.

**Keywords:** module; stability of movement; ploughing; phase–frequency characteristic

## 1. Introduction

The developed agricultural tractors that are widely used in the world, having track and wheel modifications, need constant improvement because of the increased speed of their movement during the performance of a particular technological process, the increased width of the aggregate, rational energy saturation, a decreased soil compaction, etc.

At present, there are two main ways how to increase the efficiency of the agricultural machine-and-tractor aggregates:

–   Increasing the working width of the technological part of the aggregate;
–   Increasing the working speed of the aggregate.

It should be noted that both of these ways raise their requirements for the tractor designs. The first of these ways, i.e., increasing the working width of the machine-and-tractor aggregate, requires an obligatory increase in the mass of the traction means, i.e., of the using tractor. To implement the second way, the tractor engine power should be increased.

If we analyse the basic requirements for the implementation of the noted conditions in relation to wheeled tractors, then an increase in the mass of a wheeled tractor causes such an acute negative problem as soil over compaction. To partly solve this problem, dual or even triple tires of travelling wheels, ultra-low pressure tires or arched structures, etc., are used; however, all of these technical solutions make the wheeled tractor much more complicated and more expensive, and, nevertheless, they do not give tangible results in solving this problem. It should be noted that even when using caterpillar tractors are used, the soil compaction cannot be less than when using the tractors of the wheeled modification [1].

As the investigations show (including those carried out by us), raising the efficiency of the machine-and-tractor aggregate by increasing the engine power of the aggregating tractor is a more promising direction [2]. However, in this case, much depends on the requirements of the concept that the tractor of the wheeled modification is created.

Two concepts are known presently on the basis of which the wheeled tractors are created: (1) traction; (2) traction and energy.

In accordance with the first concept, the increase in the tractor engine power ($N_e$, kW) is accompanied by a corresponding increase in its mass ($M_t$, ton). As a result of this, the energy saturation of the tractor ($E_t$, kW·kg$^{-1}$), determined by ratio $E_t = \frac{N_b}{M_t} =$ const, remains approximately the same for almost all types and designs of the wheeled tractors.

It has been established as a result of our research that the maximum power saturation of the tractor $E_t$ at which it can fully realise the engine power $N_e$ through the generated tractive effort is equal to 0.014–0.015 kW·kg$^{-1}$ [1]. Consequently, according to this concept, almost the entire value of $N_e$ is actually realised in the traction version through the running system of the power unit (the tractor). However, in this case in our soil conditions, it is impossible to realise all the installed power of the tractor engine, because of the nonlinearity of its regulatory characteristics and the oscillatory nature of the external traction load [1].

In this regard, the prospects for the development of wheeled tractors of the traction concept are currently practically exhausted since any increase in the engine power in order to ensure the required energy saturation must be accompanied by a corresponding increase in its mass. This, as remarked above, causes negative soil compaction, increases the cost of the tractor, etc.

A more advanced direction in the development of tractors is their design in accordance with the requirements of the traction and energy concept. Its essence is as follows. While, in accordance with the need to increase the efficiency of the machine-and-tractor aggregate, the engine power of the aggregating tractor should be increased; however, the value of its mass should remain practically constant. Its insignificant growth is possible only if the requirements for the strength and safety of the tractor structure are met. For this case, the power saturation of the tractor can already be represented by such a dependence $E_t = \frac{N_b}{M_t} =$ var, which makes it possible to have its value significantly higher than 14–15 kW·t$^{-1}$. So, for example, for the power unit Steyr 8300 (St. Valentin, Austria) this figure reaches 42 (www.konedata.net).

This means that the so-called "extra" power, generated by the tractor, cannot be converted into tractive effort due to the lack of its own weight. If in this case, the weight (mass) of the tractor is "artificially" increased by simple ballasting of the tractor, then it will automatically switch from the traction-energy concept to the traction concept with all the ensuing negative consequences.

From the above, it follows that the higher the $E_t$ value, the more acute the problem of using the full power of the energy-saturated tractor. Of the many ways how to solve this problem one of the most efficient, in this study, is the modular construction (design) and operation of aggregating tractors.

Such a tractor, as a rule, consists of two modules: the power module (the wheeled tractor itself) and the technological module (the additional, drive axle).

In fact, a power module is a tractor, the energy saturation of which can be significantly higher than 15 kW·t$^{-1}$. The technological module is an axle, additionally attached to the tractor, with drive wheels, its own three-point hitch, a semitrailer, a brake system and its own power take-off shaft. The wheels of the technological module are driven by the synchronous power take-off shaft of the power module (i.e., the tractor). More perspectives are hydraulic or electric drive options for the wheels of the technological module.

Under normal conditions, the power module can be used independently as a separate tractor. However, in order to significantly increase the tractive effort when performing certain agricultural operations with higher efficiency, a technological module is connected to it, as a result of which the tractive effort of this power unit is significantly increased.

On the basis of our tests and research of modular machine-and-tractor aggregates it was confirmed that the modular principle of designing tractors has a number of advantages. From a practical and scientific point of view, these advantages can be formulated as follows:

1. If you have one tractor of a certain traction class (for example, 1.4 or 2) and a technological module for it, you can successfully do without a tractor of traction class 3. That is, instead of two conventional tractors, a manufacturer of agricultural products (for example, a farmer) only needs one tractor and one technological module for it. It is quite efficient from the economic point of view.

2. The use of a technological module as part of a modular machine-and-tractor aggregate does not increase the soil compaction. This is due to the fact that the wheels of the technological module are already following the compacted track, left by the wheels of the power module. The relatively light weight power module compacts the soil insignificantly [3]. Besides, there is practically no more or less significant additional soil compaction by the wheels of the technological module. Thus, a modular-type machine-and-tractor aggregate, for example, weighing 7.5 tons will compact the soil less than the conventional tractor of the same mass (weight). The reason is that the mass of a modular-type machine-and-tractor aggregate is distributed over 3 axles while the mass of a conventional tractor of the same mass is distributed only over two. Moreover, a modular 3-axle machine-and-tractor aggregate has a more efficient "multi-pass" effect than a conventional 2-axle tractor. Due to this, such an aggregate differs from the conventional tractor by lower wheel slip and lesser specific fuel consumption.

3. Owing to the technological module, the annual load of the power module (tractor) increases significantly. For the power module (tractor) of the traction class 3, according to our calculations, it can reach 1700 h. Part of the time of the year the technological module may not be used but the losses from its downtime is about 5–7 times less than similar losses from the downtime of an idle tractor.

However, in this case, the formation of the machine-and-tractor aggregates in which an increase in the engine power of an aggregating tractor without increasing its mass requires the solution of some of its specific issues, the main of which being the question associated with the fact that the more engine power is used, the more problematic its implementation through the tractive effort of the tractor. This is particularly pronounced when its energy saturation level exceeds 15 kW·t$^{-1}$ [2].

A perspective option for efficient application of tractors with a high level of energy saturation is our proposed option for creating the so-called modular traction (module draft) device [1,3,4]. In this case, the traction (draft) device consists as if of two modules: the power module and the technological module (Figure 1). A wheeled tractor with a synchronous power take-off shaft is used as an power module. The technological module is an additional axle, driven by the wheels from the synchronous power take-off shaft of the power module, that is, from the wheeled tractor.

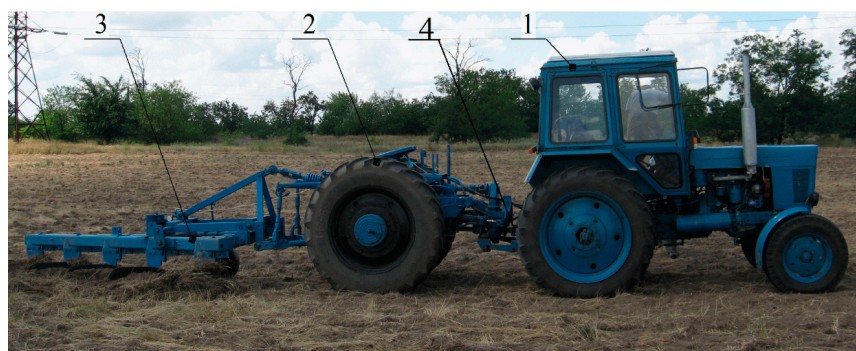

**Figure 1.** The machine-and-tractor aggregate of a modular type. 1—the power module (the wheeled tractor); 2—the technological module; 3—the aggregated agricultural machine (plough); 4—cardan transmission.

Figure 2 shows the structural implementation of connection of the power module 1 to the technological module 2 in this modular-type machine-and-tractor aggregate, which is ensured by the articulated connection 3 (vertical and horizontal pivots) and the connection of their lateral parts, using the hydraulic cylinder 4. The transmission of torque to the drive wheels of the technological module 2 is carried out from the power take-off shaft of the power module 1 through a cardan transmission.

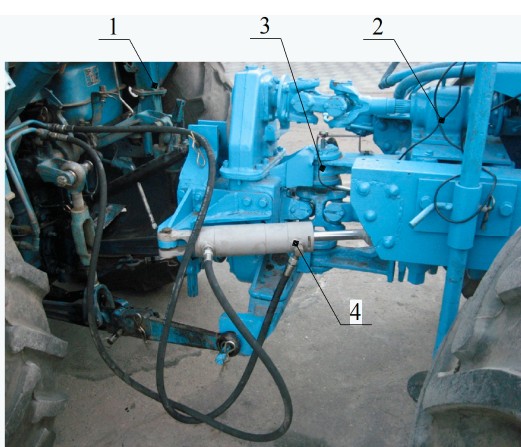

**Figure 2.** The connection of the power and the technological modules. 1—the power module; 2—the technological module; 3—the articulated joint; 4—the hydraulic cylinder, connecting the modules.

Thus, the limited turning ability of the technological module 2 relative to the power module 1 in the horizontal plane at ±30° through the above articulated joint 3 is provided by the hydraulic cylinder 4, the above-piston and the sub-piston cavities of which are interconnected. The hydraulic system of cylinder 4 is equipped with a throttle (Figure 2).

During the working movement, the external disturbances continuously act upon the technological module and the plough. These disturbances have the form of forced various amplitude and frequency fluctuations. This leads to a decrease in the stability of the movement of the technological module in a horizontal plane, which ultimately brings the power module out of the state of stable movement. All of this makes it necessary to search for ways how to increase the stability of movement, as a whole, of the entire machine-and-tractor aggregate of a modular type in a horizontal plane.

The results of investigations of the stability of movement of articulated energy devices are published in [5–13].

However, the results of previously obtained studies do not fully disclose but, most importantly, eliminate the problem under consideration, and not in all cases allow increasing the stability of movement of the modular machine-and-tractor aggregates. Therefore, we have found a new technical

solution, when the stability of movement of such a machine-and-tractor aggregate can be considerably increased, due to significant damping of the horizontal oscillatory movements of the technological module relative to the power module, due to the additional connection between them by means of a hydraulic damper. As such a damper, we proposed the use of a throttle and a hydraulic cylinder connecting the side parts of the power and technological modules. The throttle connecting the above-piston and the sub-piston cavities of the hydraulic cylinder has a resistance coefficient of $1.03 \times 10^6$ N·m·s·rad$^{-1}$ [4].

The purpose of this investigation is to theoretically and experimentally evaluate the stability of the movement of a modular-type aggregate, when the speed of its movement and the coefficient of tire slip resistance of the wheels of the technological module are changing.

## 2. Materials and Methods

Let us assume that, in reality, when performing a particular technological process, the modular aggregate performs a plane-parallel movement on the surface of the field at a constant speed $V_o$ (Figure 3). As the practice of using many agricultural aggregates shows, the assumption of the constancy of their working speed is quite correct.

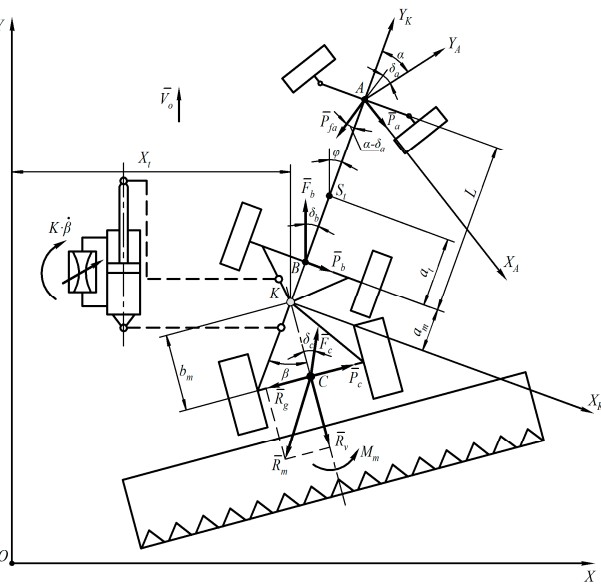

**Figure 3.** An equivalent scheme of the modular-type machine-and-tractor aggregate in its plane-parallel movement.

In order to compile a mathematical model of the plane-parallel movement of the machine-and-tractor aggregate of a modular type, performing a particular technological process, we first construct its equivalent scheme (Figure 3).

In this case, the modular-type machine-and-tractor aggregate is a multi-mass dynamic system in which the power module (the tractor) is a four-wheel model in the form of a longitudinal frame with frontal driven and rear driving wheels moving in a plane, parallel to the plane of the field surface. The technological module, which is connected to the main power module by means of a pivot, also moves in the indicated plane. The agricultural machine, performing a technological process (for example, the cultivation of crops), which is aggregated with this machine-and-tractor aggregate, is rigidly connected to the rear hitch of the technological module. This modular machine-and-tractor aggregate is shown in an arbitrary position.

Let us designate by the corresponding letters the characteristic points of the considered machine-and-tractor aggregate of the modular type: the centre of mass of the tractor—point $S_t$; the middle of the frontal driven axle is point $A$; the middle of the rear driving wheel axle—point $B$; the

point of pivotal connection of the technological module to the power module—point *K*; the point of connection of the aggregated agricultural machine (plough) to the technological module (the middle of the axle of the technological module)—point *C*.

In order to describe the plane-parallel movement of the indicated masses of the considered machine-and-tractor aggregate, which perform various types of relative displacements, it is necessary to set some coordinate systems (fixed and movable) and show them on this equivalent scheme. It is with respect to these coordinate systems that we will consider the movement of the machine-and-tractor aggregate of the modular type.

First of all, we will rigidly connect with the surface of the field the system of fixed Cartesian coordinates *XOY*, which denotes the plane in which the particular machine-and-tractor aggregate is moving. In this case axis *OX* is directed to the right in the direction of the movement of the aggregate, axis *OY* is directed in the direction of the forward movement of the aggregate (parallel to vector $\overline{V}_o$, i.e., the vector of the technologically necessary forward speed of the aggregate). Through point *K* of the articulated connection of the power and the technological modules we will draw a movable coordinate system $X_K K Y_K$ rigidly connected with the power module, in which axis $KY_K$ coincides with the longitudinal axis of the indicated module; axis $KX_K$ is directed to the right in the direction of the movement of the power module; the origin of this coordinate system is located at point *K*. The indicated coordinate system is necessary to describe the turn of the power module around point *K* relative to axis *OY*.

To indicate the direction of the movement of the frontal driven wheels of the power module, we will also choose a moving coordinate system $X_A A Y_A$ with the origin at point *A*. In addition, axis $AY_A$ coincides with the direction of the movement of the frontal driven wheels of the power module (parallel to the plane of the mentioned wheels) at an arbitrary moment, and axis $AX_A$ is directed perpendicular to axis $AY_A$ and to the right in the direction of movement of the power module. Besides, the coordinate system $X_K K Y_K$ rigidly connected with the power module, and it rotates in plane *XOY* around point *K*.

As the measurement of this turn will be the bearing angle $\varphi$, formed by the longitudinal symmetry axis $KY_K$ of the power module and axis *OY*.

In addition, during the relative movement of the aggregate in plane *XOY*, point *K* moves along axis *OX*, which is characterised by a change in the lateral displacement $X_K$ of this point. Besides that, the technological module turns around point *K* relative to the longitudinal axis $KY_K$ of the power module by a certain angle $\beta$ as a result of the action of the unfolding moment from the side of the technological module and the aggregated agricultural machine.

Thus, the machine-and-tractor aggregate of a modular type has three degrees of freedom in relation to the coordinate system *XOY*, namely: transverse displacement $X_K$ of point *K* along axis *OX*; the bearing angle $\varphi$ of the power module; the deviation angle $\beta$ of the longitudinal axis of symmetry of the technological module relative to the longitudinal axis of symmetry of the power module. The indicated coordinates are taken as generalised coordinates of the considered dynamical

$$f(x) = a_0 + \sum_{n=1}^{\infty} \left( a_n \cos \frac{n\pi x}{L} + b_n \sin \frac{n\pi x}{L} \right) \text{ system.}$$

Let us further examine all the external forces acting upon the machine-and-tractor aggregate of a modular type, and designate them in the equivalent scheme.

First of all, we will remark that for the forward movement of the considered aggregate the presence of a traction force $\overline{F}_b$, created by the rear axle of the power module, is necessary. This force is summed up from the two driving rear wheels of the power module, and it is applied at point *B*, forming a slip angle $\delta_b$ with the longitudinal axis of symmetry of the power module. Moreover, from the side of the field surface, there acts the force of rolling resistance $\overline{P}_{fa}$ of the frontal wheels of the power module, which is also summed up from the two wheels; it is applied at the point of intersection of their axis with the longitudinal axis $KY_K$ (point *A*) and is deflected from the direction of the movement of the frontal wheels by the slip angle $\delta_a$. There is also a traction force $\overline{F}_c$ of the axle of the technological module, which is composed of the two driving wheels of the technological module, it is applied at point *C* and is deflected from the longitudinal axis of the technological module by the slip angle $\delta_c$.

In addition, there are lateral forces $\overline{P}_a$, $\overline{P}_b$ and $\overline{P}_c$, composed simultaneously from two wheels, applied at points *A, B,* and *C*, respectively.

Moreover, from the side of the aggregated agricultural machine (plough) upon its working tools, there act the main moment $M_m$ and the main vector $\overline{R}_m$ of external disturbing forces.

Furthermore, vector $\overline{R}_m$ is applied at point *C* and on the equivalent scheme (Figure 3) is decomposed into the longitudinal $\overline{R}_v$ and the transverse $\overline{R}_g$ components.

The tire slips of the frontal and rear axles of the power module, as well as the axle of the technological module are expressed by angles $\delta_a$, $\delta_b$ and $\delta_c$, respectively.

As noted above, the mutual angular mobility of the technological module relative to the power module in a horizontal plane is limited by a hydraulic cylinder, the above-piston and sub-piston cavities of which are connected by a throttle having a coefficient of hydraulic resistance $K_m$. This throttle creates a hydraulic moment of resistance to the turn of the technological module relative to the power one by an angle $\beta$. The value of the indicated moment depends upon the angular turning speed $\dot{\beta}$ of the technological module, and it is equal to $K_m \cdot \dot{\beta}$, and this moment is shown in the equivalent scheme.

Let us proceed further to the compilation of differential equations of the plane-parallel movement of the machine-and-tractor aggregate of a modular type when it is performing a certain technological process.

Since the considered dynamic model is presented as a two-mass model, consisting of the mass of the power module and the mass of the technological module with an agricultural machine attached to it, and it has three degrees of freedom, it is advisable to use the method of compiling differential equations of the movement of a dynamic system, based on Lagrange equations of the second kind.

Therefore, we will further compose differential equations of the plane-parallel movement of the considered aggregate, based on the Lagrange equations of the second kind, which, as it is known, are written in the following form:

$$\frac{d}{dt}\left(\frac{\partial T}{\partial \dot{q}_i}\right) - \frac{\partial T}{\partial q_i} = Q_i, \; i = 1, 2, \ldots, n, \tag{1}$$

where *T*—the kinematic energy of the dynamic system; $q_i$—the generalised coordinate; $Q_i$—a generalised force corresponding to the generalised coordinate $q_i$; *i*—the number of the generalised coordinate; *n*—the number of generalised coordinates.

In this case, we have three generalised coordinates, $X_K$, $\varphi$ and $\beta$. Therefore it is necessary to compose three differential equations of the plane-parallel movement of the considered machine-and-tractor aggregate, corresponding to each of the indicated generalised coordinates.

The kinetic energy *T* of the indicated aggregate relative to plane *XOY* can be found from the following equation:

$$T = T_1 + T_2, \tag{2}$$

where $T_1$—the kinetic energy of the power module; $T_2$—the kinetic energy of the technological module together with the agricultural machine attached to it.

In this case, the kinetic energy $T_1$ in the plane-parallel movement of the power module will be equal to:

$$T_1 = \frac{M_{em} \cdot V_K^2}{2} + \frac{J_{em} \cdot \omega_{em}^2}{2}, \tag{3}$$

where $M_{em}$—the mass of the power module; $V_K$—the speed of forward movement of the power module (point *K*) along axis *OX*; $J_{em}$—the moment of inertia of the power module relative to point *K*; $\omega_{em}$—the angular turning speed of the power module around point *K*.

Let us express the forward speed $V_K$ and the angular speed $\omega_{em}$ in terms of generalised coordinates $X_K$ and $\varphi$ as follows:

$$V_K = \dot{X}_K \text{ and } \omega_{em} = \dot{\varphi}, \tag{4}$$

and, substituting Equation (4) into Equation (3), we finally obtain:

$$T_1 = \frac{M_{em} \cdot \dot{X}_K^2 + J_{em} \cdot \dot{\varphi}^2}{2}. \tag{5}$$

We find the kinetic energy $T_2$ of the plane-parallel movement of the technological module together with the agricultural machine attached to it from the following equation:

$$T_2 = \frac{M_{tm} \cdot V_K^2}{2} + \frac{J_{tm} \cdot \omega_{tm}^2}{2}, \tag{6}$$

where $M_{tm}$—the mass of the technological module together with the agricultural machine attached to it; $J_{tm}$—the moment of inertia of the technological module together with the agricultural machine relative to point $K$; $\omega_{tm}$—the angular turning speed of the technological module together with the agricultural machine around the point $K$.

Since:

$$\omega_{tm} = \dot{\beta}, \tag{7}$$

then, taking into account (7), from Equation (6) we obtain:

$$T_2 = \frac{M_{tm} \cdot \dot{X}_K^2 + J_{tm} \cdot \dot{\beta}^2}{2}. \tag{8}$$

After substituting (5) and (8) into Equation (2) we will have:

$$T = \frac{(M_{em} + M_{tm}) \cdot \dot{X}_K^2 + J_{em} \cdot \dot{\varphi}^2 + J_{tm} \cdot \dot{\beta}^2}{2}. \tag{9}$$

Partial derivatives of kinetic energy $T$ with respect to generalised velocities $\dot{X}_K$, $\dot{\varphi}$ and $\dot{\beta}$ corresponding to the generalised coordinates will be equal:

$$\frac{\partial T}{\partial \dot{X}_K} = (M_{em} + M_{tm}) \cdot \dot{X}_K, \tag{10}$$

$$\frac{\partial T}{\partial \dot{\varphi}} = J_{em} \cdot \dot{\varphi}, \tag{11}$$

$$\frac{\partial T}{\partial \dot{\beta}} = J_{tm} \cdot \dot{\beta}. \tag{12}$$

The time derivatives $t$ of each of the partial derivatives obtained above are determined, respectively, by the following equation:

$$\frac{d}{dt}\left(\frac{\partial T}{\partial \dot{X}_K}\right) = (M_{em} + M_{tm}) \cdot \ddot{X}_K, \tag{13}$$

$$\frac{d}{dt}\left(\frac{\partial T}{\partial \dot{\varphi}}\right) = J_{em} \cdot \ddot{\varphi}, \tag{14}$$

$$\frac{d}{dt}\left(\frac{\partial T}{\partial \dot{\beta}}\right) = J_{tm} \cdot \ddot{\beta}. \tag{15}$$

Since the kinetic energy $T$ of the dynamical system considered does not depend on the generalised coordinates, it is obvious that:

$$\frac{\partial T}{\partial X_K} = 0, \frac{\partial T}{\partial \varphi} = 0, \frac{\partial T}{\partial \beta} = 0. \tag{16}$$

Taking into account the obtained Equations (13)–(15) and (16), based on (1), we obtain a system of differential equations for the plane-parallel movement of the aggregate, based on the modular traction device, which has the following form:

$$\left.\begin{array}{rcl} (M_{em} + M_{tm}) \cdot \ddot{X}_K & = & Q_{X_K}, \\ J_{em} \cdot \ddot{\varphi} & = & Q_{\varphi}, \\ J_{tm} \cdot \ddot{\beta} & = & Q_{\beta}. \end{array}\right\} \tag{17}$$

Based on the equivalent scheme (Figure 3), we will further define the generalised forces included in the system of differential Equation (17).

In order to determine the generalised force $Q_{X_K}$ by the generalised coordinate $X_K$, we write down an expression for the elementary work of forces on a possible displacement $\delta X_K$.

$$\begin{aligned} \delta A_{X_K} & = -F_b \cdot \sin(\delta_b - \varphi) \cdot \delta X_K - P_{fa} \cdot \sin(\varphi + \alpha - \delta_a) \cdot \delta X_K + \\ & + F_c \cdot \sin(\delta_c + \varphi - \beta) \cdot \delta X_K + P_a \cdot \cos(\varphi + \alpha) \cdot \delta X_K + \\ & + P_b \cdot \cos \varphi \cdot \delta X_K + P_c \cdot \cos(\beta - \varphi) \cdot \delta X_K + R_v \cdot \sin(\beta - \varphi) \cdot \delta X_K - \\ & - R_g \cdot \cos(\beta - \varphi) \cdot \delta X_K , \end{aligned} \tag{18}$$

where $\alpha$—the turning angle of the driven wheels of the power module.

From Equation (18) we obtain that the generalised force $Q_{X_K}$ along the generalised coordinate $X_K$ will be equal to:

$$\begin{aligned} Q_{X_K} & = \frac{\delta A_{X_K}}{\delta X_K} = -F_b \cdot \sin(\delta_b - \varphi) - P_{fa} \cdot \sin(\varphi + \alpha - \delta_a) + \\ & + F_c \cdot \sin(\delta_c + \varphi - \beta) + P_a \cdot \cos(\varphi + \alpha) + P_b \cdot \cos \varphi + \\ & + P_c \cdot \cos(\beta - \varphi) + R_v \cdot \sin(\beta - \varphi) - R_g \cdot \cos(\beta - \varphi) . \end{aligned} \tag{19}$$

Consequently, this generalised force $Q_{X_K}$ is equal to the sum of the projections of all active external forces, applied to the considered aggregate, onto axis $OX$.

In order to determine the generalised force $Q_{\varphi}$ according to the generalised coordinate $\varphi$, we will write down an expression of elementary work $\delta A_{\varphi}$ on a possible displacement $\delta \varphi$. As a result, we obtain:

$$\begin{aligned} \delta A_{\varphi} & = -F_b \cdot \sin \delta_b \cdot a_m \cdot \delta\varphi - P_{fa} \cdot \sin(\alpha - \delta_a) \cdot (L + a_m) \cdot \delta\varphi + \\ & + P_a \cdot \cos \alpha \cdot (L + a_m) \cdot \delta\varphi + P_b \cdot a_m \cdot \delta\varphi. \end{aligned} \tag{20}$$

From Equation (20) we find that the generalised force $Q_{\varphi}$ along the generalised coordinate $\varphi$ will be equal to:

$$\begin{aligned} Q_{\varphi} & = \frac{\delta A_{\varphi}}{\delta \varphi} = -F_b \cdot a_m \cdot \sin \delta_b - P_{fa} \cdot (L + a_m) \cdot \sin(\alpha - \delta_a) + \\ & + P_a \cdot (L + a_m) \cdot \cos \alpha + P_b \cdot a_m. \end{aligned} \tag{21}$$

Thus, the generalised force $Q_{\varphi}$ along the generalised coordinate $\varphi$ is equal to the algebraic sum of the moments of all the active external forces acting upon the power module, relative to $K$.

And, finally, to determine the generalised force $Q_{\beta}$ by the generalised coordinate $\beta$, we will use the expression of elementary work $\delta A_{\beta}$ on possible displacement $\delta \beta$. Will have:

$$\begin{aligned} \delta A_{\beta} & = -R_g \cdot b_m \cdot \delta\beta + P_c \cdot b_m \cdot \delta\beta + F_c \cdot \sin \delta_c \cdot b_m \cdot \delta\beta + \\ & + M_m \cdot \delta\beta - K_m \cdot \dot{\beta} \cdot \delta\beta. \end{aligned} \tag{22}$$

Then, from Equation (22) we obtain that the generalised force $Q_{\beta}$ along the generalised coordinate $\beta$ will be equal to:

$$Q_{\beta} = \frac{\delta A_{\beta}}{\delta \beta} = -R_g \cdot b_m + P_c \cdot b_m + F_c \cdot \sin \delta_c \cdot b_m + M_m - K_m \cdot \dot{\beta} . \tag{23}$$

Consequently, the generalised force $Q_\beta$ along the generalised coordinate $\beta$ is equal to the algebraic sum of the moments of all active external forces acting upon the technological module together with the agricultural machine attached to it, relative to point $K$.

The constructive parameters $L$, $a_t$, $a_m$ and $b_m$, included in the previous and subsequent expressions, are shown in the equivalent scheme (Figure 3).

The obtained expressions (19), (21) and (23) for the corresponding generalised forces can be simplified since for small angles the values of the cosines can be approximately considered equal to one, and the values of the sinus are equal to the angles themselves.

Taking into account the small values of the angles included in expression (19), it can be represented as follows:

$$Q_{X_K} = -F_b \cdot (\delta_b - \varphi) - P_{fa} \cdot (\varphi + \alpha - \delta_a) + F_c \cdot (\delta_c + \varphi - \beta) + \\ + P_a + P_b + P_c + R_v \cdot (\beta - \varphi) - R_g \ . \tag{24}$$

Besides, Equation (21) can be written in the following way:

$$Q_\varphi = -F_b \cdot a_m \cdot \delta_b - P_{fa} \cdot (L + a_m) \cdot (\alpha - \delta_a) + \\ + P_a \cdot (L + a_m) + P_b \cdot a_m. \tag{25}$$

And finally Equation (23) can be presented as follows:

$$Q_\beta = -R_g \cdot b_m + P_c \cdot b_m + F_c \cdot b_m \cdot \delta_c + M_m - K_m \cdot \dot{\beta}. \tag{26}$$

The lateral forces $P_a$, $P_b$ and $P_c$, included in expressions (24)–(26), can be replaced by expressions obtained on the basis of the hypothesis of "lateral slip" of the tires of the pneumatic wheels [1]:

$$P_a = k_a \cdot \delta_a \ , \tag{27}$$

$$P_b = k_b \cdot \delta_b \ , \tag{28}$$

$$P_c = k_c \cdot \delta_c \ , \tag{29}$$

where $k_a$, $k_b$ and $k_c$—the coefficients of the lateral slip of the pneumatic tires of the running wheels of the modular machine-and-tractor aggregate.

In this case the values of the slip angles $\delta_a$, $\delta_b$ and $\delta_c$ are determined on the basis of construction of plans for the speed of points $A$ $B$ and $C$ (Figure 3) in a plane-parallel movement of the aggregate in the horizontal plane $XOY$. Using the methodology presented in [1], we built the indicated plans of speeds that made it possible to graphically find the values of speeds $V_A$, $V_B$ and $V_C$. Further, through their projections on axes $OX$ and $OY$, as well as the tangents of the corresponding angles, neglecting the small values of a higher order, the values of the slip angles themselves $\delta_a$, $\delta_b$ and $\delta_c$ were found, which are determined by the following equations:

$$\delta_a = \frac{-\dot{X}_K - (L - a_t) \cdot \dot{\varphi}}{V_o} + \varphi + \alpha \ , \tag{30}$$

$$\delta_b = \frac{-\dot{X}_K + a_t \cdot \dot{\varphi}}{V_o} + \varphi \ , \tag{31}$$

$$\delta_c = \frac{-\dot{X}_K + (a_t + a_m) \cdot \dot{\varphi} + b_m \cdot \dot{\beta}}{V_o} + \varphi - \beta \ . \tag{32}$$

By substituting Equations (30)–(32) into Equations (24)–(26) for generalized forces, and then the obtained equations into Equation (17), we produce a system of the second-order linear differential

equations, describing the plane-parallel movement an aggregate, based on a modular traction device in a horizontal plane, of the following form:

$$
\left.\begin{aligned}
(M_{em} + M_{tm}) \cdot \ddot{X}_K \;=\; & -F_b \cdot \left[\tfrac{-\dot{X}_K + a_t \cdot \dot{\varphi}}{V_o}\right] - \\
& -P_{fa}\left[\tfrac{\dot{X}_K + (L-a_t)\cdot\dot{\varphi}}{V_o}\right] + F_c \cdot \left[\tfrac{-\dot{X}_K + (a_t+a_m)\cdot\dot{\varphi} + b_m\cdot\dot{\beta}}{V_o} + 2\varphi - 2\beta\right] + \\
& +k_a \cdot \left[\tfrac{-\dot{X}_K - (L-a_t)\cdot\dot{\varphi}}{V_o} + \varphi + \alpha\right] + k_b \cdot \left[\tfrac{-\dot{X}_K + a_t\cdot\dot{\varphi}}{V_o} + \varphi\right] + \\
& +k_c \cdot \left[\tfrac{-\dot{X}_K + (a_t+a_m)\cdot\dot{\varphi} + b_m\cdot\dot{\beta}}{V_o} + \varphi - \beta\right] + R_v \cdot (\beta - \varphi) - R_g \;, \\
J_{em} \cdot \ddot{\varphi} \;=\; & -F_b \cdot a_m \cdot \left[\tfrac{-\dot{X}_K + a_t\cdot\dot{\varphi}}{V_o} + \varphi\right] - P_{fa} \cdot (L + a_m) \cdot \left[\tfrac{\dot{X}_K + (L-a_t)\cdot\dot{\varphi}}{V_o} - \varphi\right] + \\
& +k_a \cdot (L + a_m) \cdot \left[\tfrac{-\dot{X}_K - (L-a_t)\cdot\dot{\varphi}}{V_o} + \varphi + \alpha\right] + k_b \cdot a_m \cdot \left[\tfrac{-\dot{X}_K + a_t\cdot\dot{\varphi}}{V_o} + \varphi\right] , \\
J_{tm} \cdot \ddot{\beta} \;=\; & -R_g \cdot b_m + k_c \cdot b_m \cdot \left[\tfrac{-\dot{X}_K + (a_t+a_m)\cdot\dot{\varphi} + b_m\cdot\dot{\beta}}{V_o} + \varphi - \beta\right] + \\
& +F_c \cdot b_m \cdot \left[\tfrac{-\dot{X}_K + (a_t+a_m)\cdot\dot{\varphi} + b_m\cdot\dot{\beta}}{V_o} + \varphi - \beta\right] + M_m - k_m \cdot \dot{\beta} \;.
\end{aligned}\right\} \quad (33)
$$

Let us group the terms of the system of the differential Equation (33) containing derivatives $\dot{X}_K$, $\dot{\varphi}$, $\dot{\beta}$ and variables $\varphi$, $\beta$, respectively, as a result of which the indicated system of equations assumes the following form:

$$
\left.\begin{aligned}
(M_{em} + M_{tm}) \cdot \ddot{X}_K + & \left(\tfrac{k_a + k_b + k_c + P_{fa} + F_c - F_b}{V_o}\right) \cdot \dot{X}_K + \\
& + \left[\tfrac{(k_a + P_{fa})\cdot(L - a_t) - (k_c + F_c)\cdot(a_t + a_m) - (k_b - F_b)\cdot a_t}{V_o}\right] \cdot \dot{\varphi} + \\
& + (R_v - k_a - k_b - k_c - 2F_c) \cdot \varphi + \left[-\tfrac{(k_c + F_c)\cdot b_m}{V_o}\right] \cdot \dot{\beta} + (k_c + 2F_c - R_v) \cdot \beta = \\
& = k_a \cdot \alpha - R_g \;, \\
J_{em} \cdot \ddot{\varphi} + & \left[\tfrac{(k_a + P_{fa})\cdot(L + a_m) + (k_b - F_b)\cdot a_m}{V_o}\right] \cdot \dot{X}_K + \\
& + \left[\tfrac{(k_a + P_{fa})\cdot(L + a_m)\cdot(L - a_t) + (F_b - k_b)\cdot a_m \cdot a_t}{V_o}\right] \cdot \dot{\varphi} + \\
& + \left[-(k_a + P_{fa})\cdot(L + a_m) - (k_b - F_b)\cdot a_m\right] \cdot \varphi = k_a \cdot (L + a_m) \cdot \alpha \;, \\
J_{tm} \cdot \ddot{\beta} + & \left[\tfrac{(k_c + F_c)\cdot b_m}{V_o}\right] \cdot \dot{X}_K + \left[-\tfrac{(k_c + F_c)\cdot(a_t + a_m)\cdot b_m}{V_o}\right] \cdot \dot{\varphi} + \\
& + \left[-(k_c + F_c)\cdot b_m\right] \cdot \varphi + \left[K_m - \tfrac{(k_c + F_c)\cdot b_m^2}{V_o}\right] \cdot \dot{\beta} + \\
& + (k_c + F_c)\cdot b_m \cdot \beta = M_m - R_g \cdot b_m \;.
\end{aligned}\right\} \quad (34)
$$

Introducing the notation indicated below, we write the system of the differential Equation (34) in the following way:

$$
\left.\begin{aligned}
A_{11} \cdot \ddot{X}_K + A_{12} \cdot \dot{X}_K + A_{13} \cdot \dot{\varphi} + A_{14} \cdot \varphi + A_{15} \cdot \dot{\beta} + A_{16} \cdot \beta \;=\;& B_{11} \cdot \alpha - R_g, \\
A_{21} \cdot \ddot{\varphi} + A_{22} \cdot \dot{X}_K + A_{23} \cdot \dot{\varphi} + A_{24} \cdot \varphi \;=\;& B_{21} \cdot \alpha, \\
A_{31} \cdot \ddot{\beta} + A_{32} \cdot \dot{X}_K + A_{33} \cdot \dot{\varphi} + A_{34} \cdot \varphi + A_{35} \cdot \dot{\beta} + A_{36} \cdot \beta \;=\;& M_o
\end{aligned}\right\} \quad (35)
$$

where

$$
A_{11} \;=\; M_{em} + M_{tm},
$$

$$
A_{12} \;=\; \left(k_a + k_b + k_c + P_{fa} + F_c - F_b\right) \cdot (V_o)^{-1},
$$

$$
A_{21} \;=\; J_{em},
$$

$$A_{13} = \left[\left(k_a + P_{fa}\right) \cdot (L - a_t) - (k_c + F_c) \cdot (a_t + a_m) - \right.$$
$$\left. - (k_b - F_b) \cdot a_t\right] \cdot (V_o)^{-1},$$

$$A_{14} = R_v - k_a - k_b - k_c - 2F_c,$$

$$A_{15} = [-(k_c + F_c) \cdot b_m] \cdot (V_o)^{-1},$$

$$A_{16} = k_c + 2F_c - R_v,$$

$$A_{22} = \left[\left(k_a + P_{fa}\right) \cdot (L + a_m) + \right.$$
$$\left. + (k_b - F_b) \cdot a_m\right] \cdot (V_o)^{-1},$$

$$A_{31} = J_{tm},$$

$$A_{23} = \left[\left(k_a + P_{fa}\right) \cdot (L + a_m) \cdot (L - a_t) + \right.$$
$$\left. + (F_b - k_b)a_m \cdot a_t\right] \cdot (V_o)^{-1},$$

$$B_{11} = k_a,$$

$$A_{24} = -\left(k_a + P_{fa}\right) \cdot (L + a_m) - (k_b - F_b) \cdot a_m,$$

$$B_{21} = k_a \cdot (L + a_m),$$

$$A_{32} = [(k_c + F_c) \cdot b_m] \cdot (V_o)^{-1},$$

$$A_{34} = -(k_c + F_c) \cdot b_m,$$

$$A_{33} = [-(k_c + F_c) \cdot (a_t + a_m) \cdot b_m] \cdot (V_o)^{-1}$$

$$A_{36} = (k_c + F_c) \cdot b_m,$$

$$A_{35} = K_m - (k_c + F_c) \cdot b_m^2 \cdot (V_o)^{-1},$$

$$M_o = M_m - R_g \cdot b_m.$$

The system of differential Equation (35) is a mathematical model of a plane-parallel movement of the machine-and-tractor aggregate of a modular type during the execution of the technological process.

The input variables of the system of differential Equation (35) are:

1. The control impact in the form of the turning angle $\alpha$ of the driven wheels of the power module of the machine-and-tractor aggregate of a modular type;
2. The disturbing impact in the form of a summary unfolding moment $M_o = M_m - R_g \cdot b_m$.

The output parameters of functioning of the dynamic system considered are: displacement $X_K$ of point $K$— the reduced centre of mass of the modular machine-and-tractor aggregate; the bearing angle $\varphi$ of the power module; the turning angle $\beta$ of the technological module relative to the power module.

It is best to perform the analysis of the stability of the movement of the considered dynamic system using the amplitude–frequency and the phase–frequency characteristics. Such characteristics are currently used to effectively solve similar problems [1,5,11,13,14]. As it turns out, it is these characteristics that best reflect the stability of a dynamic system in the form of its response to the input disturbance.

It should be emphasized that the amplitude–frequency characteristic is the frequency distribution of the gain ratio of the input action by the dynamic system. The phase–frequency characteristic of a dynamic system is the distribution of the delay frequencies of its response to the input action, expressed by angle or by time.

For tracking dynamical systems (and the one we are considering refers precisely to those), there are ideal amplitude and phase frequency characteristics. By the way, the gain ratio by the dynamic system of the input disturbance impact (i.e., the amplitude–frequency characteristic) in the entire

frequency range should be equal to 0 [15]. The delay in the response of a dynamic system to such an impact should be as large as possible, ideally, to tend to infinity [15].

With this approach, the essence of mathematical modelling of the stability of movement of a particular dynamic system is reduced to the selection of such parameters that provide the closest approximation of the actual amplitude–frequency characteristics and phase–frequency characteristics to the ideal ones.

The actual frequency response and the phase response are obtained from the respective transfer functions. In our case the transfer function by the unfolding moment $M_o$ relative to the bearing angle $\varphi$ of the power module has the following form:

$$W_1 \; = \; \frac{b_6 \cdot p + b_5}{a_6 \cdot p^4 + a_5 \cdot p^3 + a_4 \cdot p^2 + a_3 \cdot p + a_2} \, . \tag{36}$$

A similar function by the same disturbing impact (i.e., moment $M_o$) but relative to the turning angle $\beta$ of the technological module is more complex and has the following form:

$$W_2 \; = \; \frac{b_4 \cdot p^3 + b_3 \cdot p^2 + b_2 \cdot p + b_1}{a_6 \cdot p^4 + a_5 \cdot p^3 + a_4 \cdot p^2 + a_3 \cdot p + a_2} \, . \tag{37}$$

In Equations (36) and (37) the following designations are adopted:

$$b_6 \; = \; A_{15}A_{22},$$

$$b_5 \; = \; A_{16}A_{22},$$

$$b_4 \; = \; A_{11}A_{21},$$

$$b_3 \; = \; A_{12}A_{21} + A_{11}A_{23},$$

$$b_2 \; = \; A_{11}A_{24} + A_{12}A_{23} - A_{13}A_{22},$$

$$b_1 \; = \; A_{12}A_{24} - A_{22}A_{14},$$

$$a_6 \; = \; A_{11}A_{21}A_{31},$$

$$a_5 \; = \; A_{12}A_{21}A_{31} + A_{11}A_{31}A_{23} + A_{11}A_{21}A_{35},$$

$$a_4 \; = \; A_{11}A_{31}A_{24} + A_{12}A_{23}A_{31} - A_{13}A_{22}A_{31} + A_{11}A_{21}A_{36} + A_{12}A_{21}A_{35} -$$
$$-A_{21}A_{32}A_{15} + A_{11}A_{23}A_{35},$$

$$a_3 \; = \; A_{12}A_{31}A_{24} - A_{22}A_{31}A_{14} + A_{12}A_{21}A_{36} + A_{11}A_{23}A_{36} - A_{21}A_{32}A_{16} +$$
$$+A_{11}A_{24}A_{35} + A_{12}A_{23}A_{35} - A_{13}A_{22}A_{35} - A_{12}A_{31}A_{24},$$

$$a_2 \; = \; A_{11}A_{24}A_{36} + A_{12}A_{23}A_{36} + A_{12}A_{24}A_{35} - A_{13}A_{22}A_{36} - A_{22}A_{14}A_{35} -$$
$$-A_{23}A_{14}A_{35} - A_{23}A_{32}A_{16} - A_{32}A_{15}A_{24}.$$

The methodology and algorithm for calculation of the actual amplitude–frequency and the phase–frequency characteristics of one or another dynamic system, when they work out both the control and the disturbing impacts, are described in detail in the book [15], and many other literary sources. However, before starting simulation, the developed mathematical model (35) must be checked for adequacy.

Verification of the mathematical simulation (35) for adequacy was carried out by comparing two amplitude–frequency characteristics. One of them is theoretical $A_t$ and calculated by using the transfer function (37); and the second, $A_e$, was obtained by us as a result of the experimental field study of the machine-and-tractor aggregate of a modular type during the process of ploughing.

The latter was determined under the field conditions in the process of ploughing with a machine-and-tractor aggregate of a modular type by means of a plough, attached to it. The characteristics of the modular machine-and-tractor aggregate to be studied are presented in Table 1.

**Table 1.** Technical characteristics of a modular-type machine-and-tractor aggregate with a plough.

| Parameter | Designation | Unit of Measurement | Value |
|---|---|---|---|
| Mass (weight) of the power module | $M_t$ | kg | 3820 |
| Longitudinal base of the tractor | $L$ | m | 2.37 |
| Mass (weight) of the technological module | $M_m$ | kg | 2500 |
| Mass (weight) of the plough PLN-5-35 | $M_p$ | kg | 800 |
| Moment of inertia of the power module | $\underline{J_t}$ | kN·m·s$^2$ | <u>15.7</u> |
| Moment of inertia of the technological module | $\overline{J_m}$ | kN·m·s$^2$ | 15.9 |
| Front wheel tires of the power module: | | 9.00R20 | |
| – width | $b_a$ | m | 0.24 |
| – diameter | $D_a$ | m | 0.95 |
| – air pressure | $\rho_a$ | MPa | 0.10 |
| – vertical load on the axle | $Q_a$ | kN | 12.70 |
| Rear wheel tires of the power module: | | 15.5R38 | |
| – width | $b_b$ | m | 0.40 |
| – diameter | $D_b$ | m | 1.57 |
| – air pressure | $\rho_b$ | MPa | 0.12 |
| – vertical load on the axle | $Q_b$ | kN | 25.30 |
| Wheel tires of the technological module: | | 16.9R38 | |
| – width | $b_c$ | m | 0.43 |
| – diameter | $D_c$ | m | 1.69 |
| – air pressure | $\rho_c$ | MPa | 0.13 |
| – vertical load on the axle | $Q_c$ | kN | 32.70 |
| Rolling resistance force of the frontal wheels of the power module | $P_{fa}$ | kN | 1.27 |
| Traction force of the rear axle of the power module | $F_b$ | kN | 10.10 |
| Rolling resistance force of the wheels of the technological module: | $P_{fc}$ | kN | 3.27 |
| Design parameters, shown in Figure 3 | $a_m$ | m | 1.22 |
| | $b_m$ | m | 1.22 |
| Resistance coefficient of the throttle of the hydraulic cylinder of the technological module | $K_m$ | N·m·s·rad$^{-1}$ | $1.03 \times 10^6$ |

The modular-type machine-and-tractor aggregate, together with a five-body plough attached to it, performed ploughing on a test site, 250 m long, at the same speed in each of the three variants of experiments. The background for the movement of this machine-and-tractor aggregate was winter wheat stubble after shallow ploughing. An electronic moisture meter MG-44 (Ukraine) with the following characteristics was used to measure soil moisture in the 0–10 cm layer with the following characteristics: the measurement range of the soil moisture 1–40%; the measurement error ± 1%; the duration of one measurement less than 3 s. In the research, weight moisture was measured, i.e., the amount of water per unit of mass of the dry soil. The time $t_a$ that this aggregate passed the testing section of 250 m was recorded using an FS-8200 electronic stopwatch (China) with a measurement accuracy of 0.1 s. Subsequently, the working speed $V_o$ of this aggregate was calculated by the formula: $V_o = 250 \cdot (t_a)^{-1}$.

In the process of experimental investigations, using an 8-channel analog-to-digital converter (made in Ukraine), the turning angle $\beta$ and the angular acceleration $\ddot{\beta}$ of the technological module of the modular-type machine-and-tractor aggregate with a five-body plough PLN-5-35, hung on it from behind, was recorded on the PC. To measure the turning angle $\beta$ of the technological module in a horizontal plane, we used an SP-3A sensor (made in Ukraine) with a linear characteristic and a nominal value of 470 Ω. This sensor was installed on the axis of a vertical pivot connecting the power and the technological modules of the modular machine-and-tractor aggregate (Figure 2, position 3).

The angular acceleration of the technological module was recorded using an MMA 7260QT accelerometer (Freescale Semiconductor, Phoenix, AZ, USA). Its main characteristics are as follows:

the signal—analog; acceleration measurement range ±1.5 g; sensitivity 300 mV·g$^{-1}$; the transmission capacity 150 Hz.

For implementations, as a result of the experimental research, the measurements of angle $\beta$ and acceleration $\ddot{\beta}$, the mean-square deviations and normalised spectral densities were calculated by means of the PC. The obtained statistical characteristics were used to determine the actual amplitude–frequency characteristic according to the following equation:

$$A_e = \frac{\sigma_\beta}{\sigma_m} \cdot \sqrt{\frac{S_\beta}{S_m}}, \tag{38}$$

where $\sigma_\beta, \sigma_m$—the mean-square deviations of the fluctuations of the turning angle $\beta$ of the technological module and the unfolding moment $M_o$ acting upon it, $S_\beta, S_m$—the normalised spectral densities of fluctuations of the turning angle $\beta$ of the technological module of the machine-and-tractor aggregate of a modular type and the disturbing moment acting upon the technological module.

The value of the latter was determined in this case from the expression:

$$M_o = J_m \cdot \ddot{\beta}, \tag{39}$$

where $J_m$—the moment of inertia of the technological module together with the mounted plough relative to the axis passing through the vertical pivot of the technological module (Figure 3, point $D$). This parameter was determined by calculation.

As already noted above, the experimental amplitude–frequency characteristic was compared with the theoretical one, calculated using the transfer Equation (37). Among its input parameters there are the coefficients of the tire slip resistance of the wheels of the machine-and-tractor aggregate of a modular type $k_a$, $k_b$ and $k_c$ are important. Their values depend on the vertical load $Q$ upon each tire, its diameter $D$, width $b$ and air pressure $\rho$ in it.

Since, for each of the axles of the modular machine-and-tractor aggregate, the following condition is met:

$$U = 0.42 \cdot \frac{Q}{\rho \cdot D^2} \cdot \sqrt{\frac{D}{b}} < 0.088, \tag{40}$$

then, the values of the tire slip resistance coefficients by analogy [16] can be calculated using the following expression:

$$k = 145\left(1.7 \cdot U - 12.7 \cdot U^2\right) \cdot \rho \cdot b^2. \tag{41}$$

The values of the tire slip resistance coefficients $k_a$, $k_b$, and $k_c$ were calculated by us according to expression (41), taking into account the data contained in Table 1. The numerical values of these coefficients obtained in this case were used as one of the analysed parameters, adopted when carrying out theoretical calculations of the developed mathematical model (35). This approach is explained as follows. Knowing the tendency of the impact of these coefficients upon the movement stability of the modules of the machine-and-tractor aggregate of a modular type, from Equation (41) it is easy to determine the required value of the air pressure in the tires of all its running wheels. As a result, this can be efficiently implemented under practical operating conditions of this modular machine-and-tractor aggregate.

The second parameter to be changed in the process of theoretical research was the working speed of the modular machine-and-tractor aggregate during the ploughing technological process. The velocity parameter $V_o$ was varied from 1 to 3 m·s$^{-1}$ (3.6–10.8 km·h$^{-1}$). A lesser value of movement is technologically ineffective, but a greater value is limited by the technical requirements for the modular machine-and-tractor aggregate.

## 3. Results and Discussion

In the process of experimental research the moisture content of the soil of the agrotechnical background in its upper layer 0–10 cm did not exceed 14.5%. A modular-type machine-and-tractor aggregate with a plough was moving at a working speed, the average value of which was 2.2 m·s$^{-1}$. It was this value of velocity $V_\mathrm{o}$ of the movement that was used to calculate by the transfer function (37) the theoretical ($A_t$) amplitude–frequency characteristic of processing of the disturbing impact (moment $M_\mathrm{o}$) by the machine-and-tractor aggregate studied.

Figure 4 shows the amplitude–frequency characteristic (AFC), which reflects the nature of the influence of the disturbing moment $M_\mathrm{o}$, expressed in N·m, on the deviation angle of the technological module with the agricultural machine $\beta$, expressed in radians (rad). In fact, this frequency response, like any other, is the distribution of the gain of the input signal over frequencies. In our case the input signal is the $M_\mathrm{o}$ moment (N·m), and the output signal is angle $\beta$ (rad). With the same dimension of the input and the output signals, this coefficient (i.e., the frequency response) is dimensionless. In our case, the frequency response has the dimension—rad·(N·m)$^{-1}$ and it is represented only in the frequency domain. Comparison of this characteristic with the experimental ($A_\mathrm{e}$) one demonstrated their satisfactory agreement: the maximum difference between the calculated and the field data does not exceed 14% (Figure 4). Moreover, this unambiguously indicates that the mathematical model (35) of the movement of the modular-type machine-and-tractor aggregate with an agricultural machine (in this case, a plough) attached to it is adequate and, therefore, reliable, and fully suitable for further theoretical research.

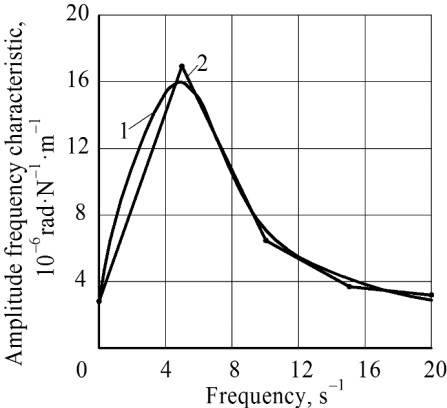

**Figure 4.** The amplitude–frequency characteristics of the investigated machine-and-tractor aggregate of a modular type. 1—theoretical $A_t$; 2—experimental $A_e$.

The further theoretical analysis performed by us of the amplitude–frequency characteristics and the phase–frequency characteristics, built on the basis of the transfer Equation (37), revealed the following. With an increase in the speed of the movement of the investigated aggregate from 1 to 3 m·s$^{-1}$, the amplitude–frequency characteristic of the turning angle $\beta$ of the technological module during the production of the disturbing impact (moment $M_\mathrm{o}$) has a resonance peak at a frequency of 5 s$^{-1}$ (Figure 5).

The amplitude frequency characteristics themselves vary insignificantly. Especially when $V_\mathrm{o} = 2$ m·s$^{-1}$, and more. This fact can be explained by the inertness of the technological module with a plough attached to it, which, when the speed of the aggregate is increased, manifests itself more effectively.

In regards to the phase–frequency characteristics, they differ also little in the frequency range 0–20 s$^{-1}$ (Figure 6).

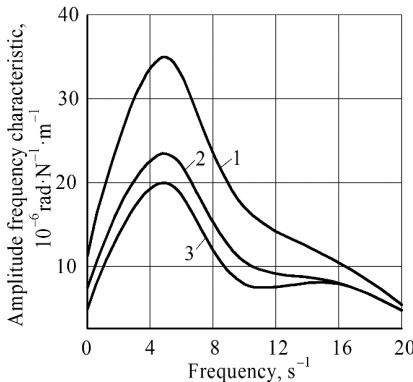

**Figure 5.** The amplitude–frequency characteristic of angle $\beta$ during the processing of the disturbing impact by the technological module at different speeds of movement of the machine-and-tractor aggregate of a modular type ($V_o$): 1—1 m·s⁻¹; 2—2 m·s⁻¹; 3—3 m·s⁻¹.

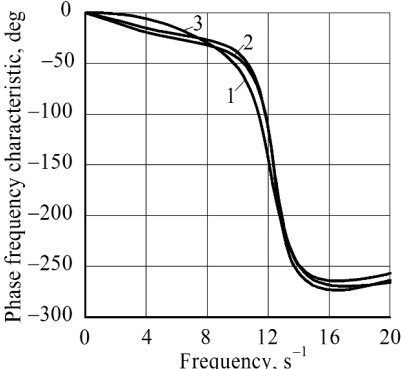

**Figure 6.** The phase–frequency characteristic of angle $\beta$ during the processing of the disturbing impact by the technological module at different speeds of movement of the machine-and-tractor aggregate of a modular type ($V_o$): 1—1 m·s⁻¹; 2—2 m·s⁻¹; 3—3 m·s⁻¹.

It should be remarked that, in processing a disturbance, the response delay (phase shift) of the dynamic system should be as large as possible. As it follows from Figure 6, the maximum phase shift of the system is approximately 270 degrees (4.7 rad), and it falls on a frequency of 16 s⁻¹. In the time calculation this means that the reaction of a modular-type machine-and-tractor aggregate to the disturbing impact will occur with a delay of 0.3 s. However, the speed of movement of this aggregate has little impact upon this process.

Hence, a conclusion can be drawn that the working speed of the movement of this modular-type machine-and-tractor aggregate in the range of its variation from 1 to 3 m·s⁻¹ has very little impact upon the process of fluctuations in the turning angle of the technological module of the modular traction device during the disturbing impart upon it in the form of an unfolding moment.

In a similar way, the same may be argued with respect to the coefficients of tire slip resistance of the frontal $k_a$ and rear $k_b$ wheels of the power module of this modular machine-and-tractor aggregate.

However, with regard to the coefficient $k_c$ of the tire slip resistance of the wheels of the technological module, there is a different result. With its increase from 160 to 210 kN·rad⁻¹, the maximum value of the amplitude–frequency characteristic of the dynamic system increases (Figure 7, Curve 2). With a further increase in the value of coefficient $k_c$, these characteristics decrease. In this case, the resonance peaks of the amplitude–frequency characteristic (Curves 3 and 4) are shifted towards higher frequencies.

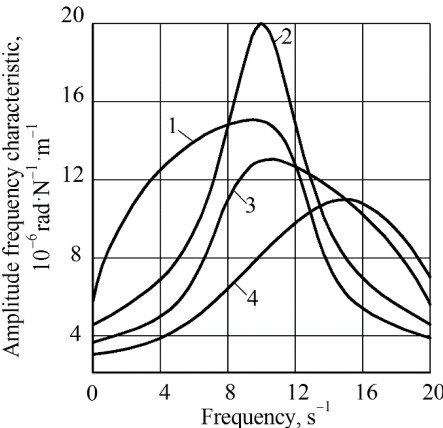

**Figure 7.** The amplitude–frequency characteristic of angle $\beta$ during the processing of the disturbance impact by the technological module for different values of coefficient $k_c$: 1—160 kN·rad$^{-1}$; 2—210 kN·rad$^{-1}$; 3—260 kN·rad$^{-1}$; 4—310 kN·rad$^{-1}$.

As for the graphs of variations in the phase–frequency characteristics, despite the fact that they differ from each other at frequencies of more than 6 s$^{-1}$, this difference is minimal.

Therefore, there is every reason to assert that the installation of wheels of the technological module with a slip resistance coefficient equal to 260 kN·rad$^{-1}$ and more helps to reduce the amplitude of its fluctuations in a horizontal plane.

Next, let us analyse the impact of the unfolding moment $M_o$ upon the dynamics of variations in the bearing angle $\varphi$ of the power module of the machine-and-tractor aggregate of a modular type. In this case the transfer function, defined by Equation (36), can be used to build the amplitude phase frequency characteristics.

As in the variant with the technological module, a change in the speed of the movement of the machine-and-tractor aggregate of a modular type in the considered speed range 1–3 m·s$^{-1}$ affects insignificantly the nature of processing of the disturbing impact by the power module in the form of fluctuations of the $M_o$ moment (Figure 8). Here, as in the case of the technological module, the resonance peaks of the amplitude–frequency characteristic fall at a frequency of 5 s$^{-1}$.

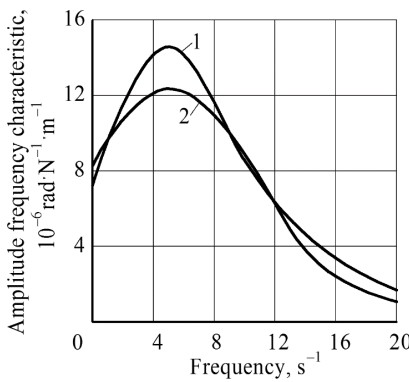

**Figure 8.** The amplitude–frequency characteristic of angle $\varphi$ during the processing of the disturbing impact at different values of speed $V_o$: 1—1 m·s$^{-1}$; 2—3 m·s$^{-1}$.

However, the amplitude of fluctuations of angle $\varphi$ is considerably less in comparison with the amplitude of fluctuations of angle $\beta$. So, at a speed of this aggregate, equal to 1 m·s$^{-1}$, the maximum value of the amplitude–frequency characteristic of fluctuations in the bearing angle of the power module (Figure 8, Curve 1) is about 2.4 times less than the same value of the amplitude–frequency characteristic of fluctuations in the turning angle of the technological module (Figure 5, Curve 1). At a speed of the movement of the machine-and-tractor aggregate of a modular type, equal to 3 m·s$^{-1}$,

the amplitude of fluctuations of angle $\varphi$, in comparison with the fluctuations of angle $\beta$, decreases 1.6 times.

The obtained result may be considered quite logical since the mass of the power module of the machine-and-tractor aggregate of the modular type is 520 kg more than the mass of the technological module together with the plough attached to it. Moreover, the unfolding moment $M_o$ acts directly upon the technological module with the plough but not on the power module.

It is this circumstance that can also explain the fact that the change in the values of the tire slip resistance coefficients of the wheels of the rear and, especially the frontal axles of the power module, has weak influence upon the fluctuations of its bearing angle under the impact of the disturbing moment. This is quite logical since the value of this coefficient is most dependent on the air pressure in the tire [16]. Hence, it follows that, in case of choosing this parameter (i.e., $\rho$), one should take into account to a greater extent not the stability of the power module against the action of the disturbing impact but other factors.

It should be remarked that, the retardation dynamics of the response of the power module to the fluctuations of the disturbing moment $M_o$, is such that, when the disturbance fluctuates with a frequency of up to $8.5\,\mathrm{s}^{-1}$, a change in the speed of the aggregate from 1 to $3\,\mathrm{m\cdot s}^{-1}$ has little impact upon the fluctuation dynamics of the bearing angle of the power module of the modular machine-and-tractor aggregate (Figure 9).

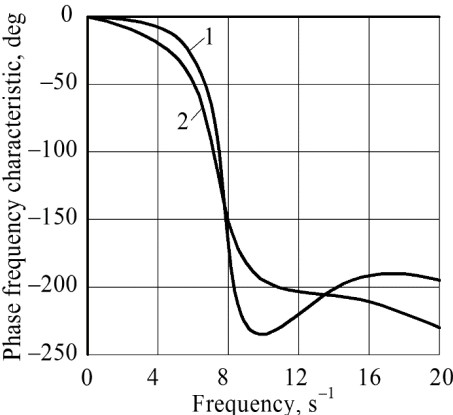

**Figure 9.** The phase–frequency characteristic of angle $\varphi$ during the processing of the disturbing impact at various values of speed $V_o$: 1—$1\,\mathrm{m\cdot s}^{-1}$; 2—$3\,\mathrm{m\cdot s}^{-1}$.

This also gives grounds to assert that in the range of fluctuations of the disturbing moment of $8$–$14\,\mathrm{s}^{-1}$ it is preferable for the machine-and-tractor aggregate of a modular type to move at a lower speed since in this case the delay in the reaction of the power module to the disturbing impact is large. Especially at the disturbance fluctuation frequencies, close to $10\,\mathrm{s}^{-1}$ (Figure 9, Curve 1).

At fluctuation frequencies of the disturbing moment more than $14\,\mathrm{s}^{-1}$ it is preferable for the investigated modular-type machine-and-tractor aggregate to move at a speed of up to $3\,\mathrm{m\cdot s}^{-1}$ (Figure 9, Curve 2). In this case, especially at frequencies greater than $15\,\mathrm{s}^{-1}$, the desired delay in the response of the power module of the modular machine-and-tractor aggregate to the disturbing impact increases by more than 0.5 rad.

Thus, we have obtained the basic kinematic and design parameters of a modular-type machine-and-tractor aggregate that ensure its stable movement during the execution of the technological process, in this case, ploughing.

## 4. Conclusions

The developed new mathematical model of the movement of a modular-type machine-and-tractor aggregate during the ploughing process is adequate. The research results obtained on the basis of its use show the following:

- A change in the working speed of this aggregate during ploughing from 1.0 to 3.0 m·s$^{-1}$ does not lead to the deterioration in the stability of the movement of either the technological or, all the more, the power module of the machine-and-tractor aggregate of a modular type. The amplitude–frequency characteristic of processing the disturbing unfolding moment by them when parameter $V_o$ is increased, although insignificantly, improves. The phase–frequency processing of the technological module of the modular-type machine-and-tractor aggregate somewhat deteriorates but only at relatively high frequencies of its fluctuations—more than 10 s$^{-1}$. The delay in the reaction of the power module of the machine-and-tractor aggregate of a modular type is practically invariant with respect to the change in the mode of the movement of this aggregate in the range 1–3 m·s$^{-1}$.

- The values of the coefficients of resistance to tire slip of the wheels of the power module do not have a noticeable impact upon the processing of fluctuations of the disturbing moment. At the same time, the value of the coefficient of resistance to tire slip of each wheel of the technological module of the machine-and-tractor aggregate of a modular type must be not less than 160 kN·rad$^{-1}$.

- As a result, equipment of the technological module with tires, with a withdrawal resistance coefficient of at least 160 kN·rad$^{-1}$, along with the installation in its hydraulic cylinder of a throttle, with a resistance coefficient of $1.03 \cdot 10^6$ N·m·s·rad$^{-1}$ allows to ensure stability of the working movement of the aggregate, based on the modular means, in the range of speeds from 1 to 3 m·s$^{-1}$.

**Author Contributions:** Conceptualization, V.B. and S.I.: methodology, V.B., A.A., and V.N.; software, I.H. and V.N.; validation, V.B., A.A., and I.H.; formal analysis, I.H.; investigation, V.B and S.I.; resources, V.B.; data curation, V.N.; writing—original draft preparation, I.H. and V.N.; writing—review and editing, V.B., S.I., and A.A.; visualization, V.N.; supervision, V.N.; project administration, V.B.; funding acquisition, H.B. All authors have read and agreed to the published version of the manuscript.

**Funding:** This research received no external funding.

**Conflicts of Interest:** The authors declare no conflict of interest.

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
