# Peer review of "A Mathematical Model of Plane-Parallel Movement of the Tractor Aggregate Modular Type"

_agriculture, doi:10.3390/agriculture10100454_

Round 1

Reviewer 1 Report

General remarks

It’s a real engineering paper (e.g. formulation the purpose of this study is to increase the stability of movement) outlining a well-known issue of converting the vehicle mass into draft power for operating agricultural implements effectively.

In the introduction this issue is discussed with high expertise but without any references. The machine configuration the authors propose is unconventional but logical.  The approach is a multi-axle tractor, which is a frequently discussed concept. Current knowledge is that multi passed tracks result in many experiments in deeper penetration of tension in the soil.

The major part of the paper deals with the dynamic of the machine concept and even limited to the assessment of stability in the horizontal plane.

Low care is taken on international scientific standards as the structure of the paper should present an introduction with references. 

The main agronomical point of the study is the increase of traction efficiency of a multi-axle tractor operating a five-body plough. Results related to this central topic are presented indirectly only.

The chapter on development of the model is of low interest for a reader from the field of agronomy. For page 7 to 16 needs special expertise in modelling which suggests to share the paper into modelling part which is a special field of expertise and a more agronomy related paper presenting results about the progress of efficiency of the proposed aggregated vehicle.

English wording needs to be improved, many phrases are not understandable

Keywords. repeating words and terms already given in the title is undesirable

Introduction

Line

44 -46 please, formulate shorter sentences which are understandable

56 i.e. of the aggregating tractor; can not understand what is meant

 64 using used

65 is there a reference for this statement? Same for 66

74 a formula needs to be indexed

71 please use international units m for mass in g or kg, power is P

77 this general statement is not true for all tires and soil conditions

94 in a scientific paper a sound specification of the employed technical device and instrumentation should be given, using the brand name to specify the technical equipment is not adequate.

122 and 126 are there results on dynamic vertical wheel forces or on soil compaction giving evidence for these statements?

139 4. Statement is not clear, rephrase to make it understandable

152 this reads like a company’s advertising leaflet with a pinch self-praise, please rephrase in a more neutral statement and make clear what is an efficient connection.

Latest from these lines the text is matter of methods and material

165 fig 2 please indicate the device for power transmission

159 repetition, has been explained before

169 limited turning ability please give angle

171  pistons are not this visible in fig 2

172 is controlled by a throttle?   

173 to 175 never ending sentence please rephrase to make it understandable

196 tire slip resistance coefficients, please explain what is the difference to slip which is a relative unit.

M&M

483 soil water content: volumetric as per dry matter?

498 Om ?

Results

Fig 4/ 554  Amplitude frequency characteristic in rad per Nm of which variable?

How it would look like in time domain?

The reader of an agricultural journal gets not informed about the progress of the driving cart (technological module) but learn about the horizontal stability of the vehicle configuration when employed for ploughing. The horizontal torque originates by lateral forces of the plough. How are the plough bodies are equipped? The magnitude of the torques and forces do not appear in the paper.

Conclusion

There is no result on increase efficiency using aggregated tractor-cart system.

674 this statement  can not be concluded from the results in the figures.

Reviewer 2 Report

The aim of this work is the improving of the stability of the movement of the machine-and-tractor aggregate of a modular type by performing theoretical and experimental research changing in the speed of its working movement, as well as in the tire slip resistance coefficient of the wheels of the technological module. The output parameters of the model are the turning angle of the technological module relative to the power module in the horizontal plane and the course angle of the latter.

However, in the description of experimental research, it misses details of the general approach and its applicability. In addition, it is a very good study, but I would propose a more detailed contextualization and a more critical result discussion.

Major comments

Title

Title too long and difficult to understand.

Abstract

The abstract is too long (max 200 words from the author guidelines).

Keywords

The same words as the title must not be used as keywords.

Introduction

Generally, the introduction is not contextualized with respect to the aims and scopes of the journal (otherwise change it). I would add the benefits that such a mathematical model can bring to agriculture that is constantly evolving (e.g., precision agriculture, use of new technologies, etc.).

In addition, there are too few bibliographical references. This does not make possible comparison with the current literature.

L58-65: it is strictly necessary to investigate the problem of soil compaction and workability in agriculture, otherwise the purpose of the work is not well introduced.

L102: (but in general throughout the text) modify “in our opinion” with “in this study” (in scientific texts it is better to keep the impersonal).

L182-182: generally, it is not enough to insert the bibliographic references, but for these, it is important to critically add the results of the mentioned works. This is just an example, there are many others in the whole document.

The aim of this work is too generic and not very explicit. Expand and define it better.

Theoretical premises

L198: modified “Let us…”

Materials and Methods

From L482: the brands of the instruments mentioned should be indicated.

Results and discussion

In general, this section is very well organized, but I would contextualize the results comparing them with other present in literature.

Round 2

Reviewer 1 Report

Thank you for following and understanding my comments 

Reviewer 2 Report

The manuscript in the present form can be published in Agriculture as the authors have been partially but sufficiently improved it. Parts that better contextualize the work have been not added but only citated. However, the fact of not adding some required details has been justified. The aim is now clearer.

I don’t need to review another version because I accept the work in the present form.

Best regards